# Out of Africa: Juvenile Dispersal of Black-Shouldered Kites in the Emerging European Population

**DOI:** 10.3390/ani12162070

**Published:** 2022-08-14

**Authors:** Domingo Rivera, Javier Balbontín, Sergio Pérez Gil, José María Abad Gómez-Pantoja, Juan José Negro

**Affiliations:** 1GPEX, Área de Trabajos en el Medio Natural, Edificio Tercer Milenio, Avda. de Valhondo s/n 06800 Mérida, 06006 Badajoz, Spain; 2Departamento de Zoología, Facultad de Biología, Universidad de Sevilla, Edificio Verde, Avda. de Reina Mercedes s/n, 41012 Sevilla, Spain; 3GEACAM, Gestión Ambiental de Castilla la Mancha, c/Escalona 89, 3ºK, 28024 Madrid, Spain; 4Departamento de Anatomía, Biología Celular y Zoología, Facultad de Biología, Universidad de Extremadura, Avda. de Elvas s/n, 06006 Badajoz, Spain; 5Department of Evolutionary Ecology, Estación Biológica de Doñana (CSIC), Avda. Americo Vespucio 26, 41092 Sevilla, Spain

**Keywords:** brood rank order, natal dispersal, laying date, range expansion, resource competition hypothesis, wandering hypothesis

## Abstract

**Simple Summary:**

Black-shouldered Kite (*Elanus caeruleus*) is a relative recent colonizer in Europe. The study of juvenile dispersal may help to better understand the patterns of range expansion and colonization. In this study, we provide some information about patterns of juvenile dispersal according to sex, habitat quality, timing of reproduction, and nesting hatching order to test two competing hypotheses about natal dispersal in this poorly studied raptor. We found some evidence supporting the Resources Competition Hypothesis since nestlings hatched from high quality territories and hatched first within the brood stayed closer from natal areas than nestlings hatched from low quality territories or later hatched nestlings.

**Abstract:**

Knowledge of animal dispersal patterns is of great importance for the conservation and maintenance of natural populations. We here analyze juvenile dispersal of the poorly studied Black-shouldered Kite *(**Elanus caeruleus*) monitored in southwestern Spain in an ongoing long-term study initiated in 2003. The European population of Black-shouldered kites is thought to be a recent one funded by colonizing African birds, as no kites have been found in the European fossil record, and the breeding population has progressively expanded to the North in the late 20th and 21st centuries. We obtained information on movements behavior during dispersal from 47 juveniles Kites after marking 384 nestlings with wing tags and three nestlings with radio transmitter. We have tested two competing hypotheses (i.e., the Resources Competition Hypothesis and the Wandering Hypothesis (WH)) that may explain the leptokurtic distribution of the natal dispersal distance in *Elanus*. After independence, juvenile females dispersed farther from the natal areas than males, as is common in birds. On average, males and females dispersed from their natal areas over 9 (i.e., 26.15 km) and 15 (i.e., 43.79 km) breeding territories, respectively. A male and two females dispersed further than 100 km from their natal nest. Our results indicated some evidence supporting the competition-for-resources hypotheses since nestlings hatched from high quality territories stayed closer from natal areas than nestlings hatched from low quality territories and also nestlings hatched first within the brood also tend to recruit closer to their natal area than later hatched nestlings which tend to disperse further away from their natal area. The information provided by these crucial demographic parameters will be used for the elaboration of future conservation plans for the management of this colonizing species in Europe.

## 1. Introduction

One of the characteristics that best defines animals is their ability to move. The movement of individuals between different habitat patches allows the transfer of genes in heterogeneous environments, which results in healthy populations having enough genetic variability to resist environmental challenges [1,2]. The most crucial movement in terms of gene flow is natal dispersal, which is defined as the movement that a certain individual does between its place of birth and the place or territory where it starts reproduction for the first time [3]. In birds and other taxa, this period comprises three phases, the first one beginning once the ties with the parents and the home natal territory are broken, which corresponds to an emigration stage. In a second phase, the animals settle and wander in areas with high food availability and scarcity of competitors (i.e., transfer stage) and a third phase in which the individual selects the area where it will make its first reproductive attempt (i.e., immigration stage) [1]. The study of juvenile dispersal has attracted the interest of ecologists and evolutionary biologists dedicated to biodiversity conservation [4]. However, the study of dispersal movement may be difficult in those species in which the movement capacity is high, and some individuals may leave the boundaries of the study area [5,6,7,8]. On many occasions, it is hard to know if marked individuals remain undetected because they have moved outside the investigators’ detection area or because they have not survived [9]. However, information gathered in local study areas is still valid since it may advance our knowledge of dispersal behavior even if limited by an uncertain bias due to long-distance events of dispersal. Accordingly, in recent years a wealth of information has been collected on dispersive movements in a variety of taxa, including birds, mammals, reptiles, fish, and some invertebrates [10,11,12,13,14,15,16].

These studies show that natal dispersal distances have a leptokurtic distribution, with most individuals showing short natal dispersal distances and a few moving a greater distance from their place of birth [13,17,18,19,20,21]. Familiarity with the area of birth and the attraction of conspecifics would explain why there are a higher number of individuals that recruited into their breeding population in areas close to their natal area [22,23,24,25,26]. There are two conflicting explanations for individuals who travel farther and have greater natal dispersal distances. The hypothesis of Competition for Resources or mates (hereafter, CRH) suggests that individuals in the right tail of the distribution are less competitive and are displaced by other more competitive individuals that stay closer to the natal area [27,28,29,30,31]. On the contrary, the “Wandering Hypothesis” (hereafter, WH) suggests that individuals in the right tail of the distribution (i.e., those that travel further) are individuals in better physical condition, who were born earlier during the breeding season, allowing them to have more time and energy to confront the costs associated with dispersive movements before food availability decreases at the end of the breeding cycle [18,31].

The scientific literature on dispersal also shows that natal dispersal is often sex-biased in numerous taxa [32,33,34]. For instance, in mammals, usually males disperse farther than females, whereas in birds, the sex that disperses farther is the female. This observation has traditionally been explained according to the differences in the mating system shown by these two groups of vertebrates. Thus, most mammals are polygamous with males mating with more than one female and males not contributing to the care and defense of their young. Therefore, in this case familiarity with the area is less important and males will gain fitness by dispersing further and trying to encounter as many mates as possible. On the contrary, most birds are socially monogamous, with males contributing as much as the females to the care and defense of the nestlings, and thus familiarity with the breeding area becomes more important [32,35]. Nonetheless, empirical evidence supporting that sex-biased dispersal is linked to mating system is scarce, and recent reviews have not found evidence supporting this explanation [33,36,37,38].

The aim of this 17-year-long study is to describe juvenile dispersal in the Black-shouldered Kite *(**Elanus caeruleus*) in Spain. This species is the only raptorial species added as a breeder to the European avifauna from Africa in historical times [39,40,41]. The social behavior and breeding site tenacity of African Black-shouldered Kites were studied in the southern hemisphere quite long ago [42], as the onset of breeding of the species was deemed to be unpredictable and the populations were considered irruptive following prey outbreaks, and thus different to most other diurnal raptors [43]. Recent colonization of Europe and the specific movement behavior of this species make it an attractive candidate for the study of juvenile and natal dispersal.

The occupation of breeding territories and the laying date was used as a proxy of territory quality to discriminate between the “CRH” and “WH” hypotheses. According to CRH, we expected that the dispersal distance should be inversely correlated with the quality of the natal habitat. Specifically, it was expected that those individuals that disperse further should be less competitive, and hence hatched from territories with a low occupation rate and/or from late breeding pairs. On the contrary, if the relationship between the quality of the natal habitat and juvenile dispersal distance is positive, this would support the “WH”. In this case, those individuals that disperse further should be hatched from territories with a high occupation rate and/or from early breeding pairs. In addition, those individuals that hatched first within the brood are expected to be more competitive compared to late-hatched siblings. Accordingly, it was expected that the hatched order within the brood would explain some of the variation in dispersal distance. Specifically, first-rank siblings should stay closer to their natal area compared to second-rank siblings according to CRH and the contrary according to WH.

## 2. Methods

### 2.1. The Study System

The Black-Shouldered Kite is a small bird of prey (*c.* 270 g) that occupies territories during the breeding season and defends them against intruders and competitors. The species seems to have colonized Europe form Africa quite recently, as no individuals have been identified in the European fossil record [44,45], and the species breeding range has advanced progressively to the north in the 20th century [46,47]. Egg laying takes place from the end of February to mid-May with an average laying date around mid-March in our study area in southwestern Spain. Females lay between 2 and 6 eggs, and the incubation period lasts 30–35 days. Nestlings remain in the nest for another 30–35 days before fledging [48,49]. Frequently, during the final phase of the chick-rearing period, the female leaves the territory, and the male alone provides the offspring [50]. Once the fledglings perform the first flight, they stay in the parent territory for about 40 days in a comparatively long post-fledgling dependence period, until parental ties are broken, and the dispersive period begins [51]. The age at first reproduction is about one year for most individuals, but we have observed some six-month-old females as reproductive individuals [49]. The Black-shouldered Kite is nonmigratory in Spain, and numerous individuals –sometimes about 60 may aggregate at social roosting sites at night associated with habitats with high prey availability in the winter months [52]. They predominantly hunt small rodents and could prey more occasionally on passerine birds, insects, and reptiles [53].

### 2.2. Field Procedures

Breeding pairs of Black-shouldered Kites were monitored in approximately 2.800 km^2^ located in southwestern Spain every year from 2003 to 2020, making it the longest lasting study for any *Elanus* population in the world. The study area is an agricultural mosaic of both no irrigated and irrigated fields located on primarily flat relief in the Guadiana River basin. Some of these fields are occupied by oak trees (i.e., *Quercus* spp.) dispersed within cereal fields, which is the preferred habitat for nesting of Black-shouldered Kites. The nearest neighbor distance (NND) in the study area was 2.850 m (for details, see [54]. From January to July we searched the study area looking for signs of territorial pairs (i.e., territorial displays and delivery of nest material). The territories were considered active when the female laid eggs and incubated. We made at least three visits to the territories, one to locate the breeding pair, another during the incubation period, and a third visit when the chicks were 15–30 days old. The mean ± SD number of breeding pairs monitored was 27.05 ± 20.29 (range: 2–70). The nestlings were marked in the nest and the tarsal measurements were taken with a Mitutoyo digital caliper with an error of ±0.01 mm and weighed with a Pesola Spring scale to the nearest 1 g. A feather was taken from the back for molecular sexing following the methodology of [55]. Some of the nestlings were marked with wing tags with an alphanumeric binomial and a color code. The tags were placed on the patagium on the right wing when the plumage had an adequate development and it was certain that it would not harm the nestlings (Figure 1a). In total, 562 nestlings have been marked with metal rings and 384 of them were also provided with wing tags. In addition, 3 nestlings were tagged with backpack VHF radio transmitters (Biotrack 3–4 g) attached with a Teflon ribbon and a cotton weak link in the sternum (Figure 1). This study was finally based mainly on resighting of wing-tagged birds (*n* = 44 individuals) and only three recoveries were based on radio transmitters birds. Hence the total sample size for the study of juvenile dispersal was 47 individuals. From the observations based on radio-transmitter birds, two of these were based on birds found dead and the other was located 2.8 km from their natal sites and was not a long-distance dispersing individual. Therefore, there was no difference between wing tagged and radio transmitter birds on juvenile dispersal distance based on the type of marking.

### 2.3. Dispersal Measurements

The beginning of the dispersal period was established when parental ties were broken (personal observation), or when the offspring was detected outside of the parental territory, that is at a distance greater than NND (i.e., 2850 m), and the age of the offspring was older than 75 days old. Except for one individual that was found dead at 87 km from their natal site at the age of 65 days and was far enough from their nest to considerer this individual emancipated from parental ties. The mean age of the resighting birds was 420 days (SD = 491 days) (range: 65–2420 days). Because the age of first reproduction is at the age of one year and there are records of two birds reproducing at the age of six months, the maximum distance recorded on this study should be highly correlated with natal dispersal distance. However, to test the hypotheses explaining variation in natal dispersal, we used a sub-sample of 22 individuals that were territorial (i.e., occupying a breeding territory for their first time upon wing-tagged lecture) and accurately reflecting natal dispersal. Juvenile dispersal distance was measured as the Euclidean distance between the site of birth and the farthest site in which the individual was detected during the juvenile dispersal period. Likewise, natal dispersal distance was measured as the Euclidean distance between the site of birth and the first site the individuals were found occupying a breeding territory for the first time.

The bearing of dispersal was calculated taken as the origin point the birthplace, and as the end point the re-sighting site. For this purpose, we draw lines from the origin to the end point and used the azimuth function in QGis [56].

### 2.4. Statistical Analyses

For the “juvenile dispersal” analyses, ordinary least square (OLS) regression analyses was used to test if juvenile dispersal distance was statistical different across sexes. This analysis was based on 47 individuals observed after wing-tagged or radio-transmitter marking during the juvenile dispersal period. In this analysis, juvenile dispersal distance was the dependent variable (log-transformed prior to the analysis) and sex was included as a factor. We run complementary analyses for the “juvenile dispersal” analysis including nest site geographical coordinates to control for spatial autocorrelation. The result of this model did not change with respect of a model without geographical coordinates. For simplicity and to avoid overparameterization, we only show the model without geographical coordinates included in the statistical model.

The preferred direction (azimuth) of juvenile dispersal was studied by categorizing the calculated bearing as: North (bearings between 315°–45°), East (45°–135°), South (135°–225°), and West (225°–315°) and used the Chi-square test to determine whether there was a statistically significant difference between the expected frequencies if the direction of dispersal was random and the observed frequencies of dispersing individuals.

To explain variation in “natal dispersal” distance, we used general linear model (GLM) in which natal dispersal distance was the dependent variable. We transformed it to a logarithmic scale prior to the analysis. We included sex (factor), brood order (factor), laying date, and territory occupancy as predictor in a full model. The laying date was the Julian date of the first egg laid in the territory where the chick was tagged. The brood order was a factor of three levels (i.e., first-second-third). The occupancy of the territory was measured as the proportion of years a given territory was occupied throughout the study period. This measure is considered a reliable proxy of territory quality in birds of prey [57]. This analysis was based on a sub-sample of 22 individuals that were observed already recruited as territorial for the first time on a breeding territory. Thus, it is the sub-sample on which the hypotheses were tested as described in the Introduction section. Because we have 22 observations and four predictors, we compared candidate models using stepwise selection procedure alternating backward and forward in both direction until reaching a reduced model and including therefore one predictor at a time during the selection procedure, in order to minimize the problem of the over-parametrized full model. For the same reason (i.e., low sample size) it was not suitable to test for second order effects or other possible model fit to data (e.g., using generalized additive model). Statistical difference was set at α = 0.05. We employed the MASS package using R version 4.1.2 [58].

## 3. Results

### 3.1. Juvenile Dispersal

The mean (±SD) distance (SD) from the natal nest was 26.15 km (±72.70) km (range = 1.20–404 km, *n* = 30), for males and 43.79 km (±63.59) km (range = 1.25–240 km, *n* = 17) for females. The median was 7.35 km for males and 16.61 km for females. A male and two females spread out more than 100 km from their natal nest. One of these observations was obtained from the re-sighting of a wing-tagged living individual. However, the other two records observed of long-distance dispersal (i.e., >100 km) were obtained from two individuals found dead and recovered by third parties (Figure 2). We found that 22 (46.8%), 6 (12.7%), 7 (14.9%), and 12 (25.5%) out of 47 resighting were North, East, South, and West, respectively, from their birthplace. These directions in juvenile dispersal deviated significantly to a North direction (χ^2^ = 13.41, *d.f.* = 3, *p =* 0.0038).

There was a significant effect of sex on juvenile dispersal distance (estimate (s.e.) = −0.745 (0.39), *t* = −1.99, *p* = 0.05). If the location (latitude, longitude, and the interaction latitude* longitude) was included in the same model to control for spatial autocorrelation the effect of sex on juvenile dispersal distance did not change (estimate (s.e.) = −0.793 (0.37), *t* = −1.96, *p* = 0.05). Several observations of juvenile dispersal were from individuals that share a nest site. Thus, to test if it was necessary to include a random effect for nest site on a linear mixed effect model, a linear mixed effect model with the nest identification (i.e., nest site ID) as a random intercept was compared with a linear model without the random effect of nest site ID. The result of this comparation indicated that it was not necessary to include this random intercept (ꭕ^2^ = 0.77, *d.f.* = 1, *p* = 0.60). The year when the nestlings were banded did not explain too much variance in juvenile dispersal distance either. Thus, a linear mixed effect model with year as a random intercept compared with a linear model without the random effect of year indicated that year was not significant statistically (ꭕ^2^ = 0.024, *d.f.* = 1, *p* = 0.87). Thus, the effect of year was not included. Moreover, including year as a random (intercept) and nest site ID (intercept) on a linear mixed effect model with latitude and longitude to control for spatial autocorrelation and the same predictors as a GLM did not change the results. Therefore, there was a difference between male and female on the distance travelled from the natal nests during the juvenile dispersal period. Thus, males stay closer to their natal nests than females, which disperse further (Figure 3).

### 3.2. Natal Dispersal

In the analyses of natal dispersal distance, it was found that those nestlings raised from parents that occupied territories with a higher occupation rate stayed close to their natal areas than nestlings raised from parents that occupied territories with a lower occupation rate (Figure 4). Furthermore, although not significant, a trend was also found in that the first order nestlings within the brood tend to stay closer to their natal nest compared to nestlings hatched second or third within their brood (Figure 5). We did not find a significant effect of laying date nor of sex on natal dispersal distance (Table 1).

Order is a factor with three levels (first–second–third). Territory occupancy is the percentage of years that a given territory has been occupied by territorial breeding pairs which is a measure of territory quality in raptors [57].

## 4. Discussion

Our results showed evidence supporting the resources for competition hypothesis since nestlings hatched in good quality territories disperse shorter distances than nestlings hatched in low quality territories when using occupation territory rate as a proxy of territory quality. A near significant effect of brood rank order on natal dispersal distance was also found further supporting the resources for competition hypothesis. Although during the juvenile dispersal period females moved further than males from natal areas, differences in natal dispersal distance between male and female was not found for a smaller sub-sample of individuals that were already recruited to their first breeding territory and hence natal dispersal distance could be analyzed. However, it should be noted that these results come from a low sample of individuals and future studies will be needed using a larger sample size and new predictors to try to discern between these competing hypotheses.

The movement of individuals during their lifetime allows for the interchange of genetic material that is crucial for the maintenance of outbred natural populations [3,59]. In terms of breeding territories based on NND, males and females dispersed from their natal areas on average over 9 and 15 breeding territories, respectively. These distances were along the range of distance found for other birds of prey of similar size [30,60,61]. Nevertheless, it is important to highlight that this study was mainly based on wing-tagged individuals, and therefore it probably underestimated long-distance dispersal records [9]. However, this is a valuable study that provides important information on juvenile dispersal in a poorly studied raptor that thrives in low numbers along the European distribution range [62].

These results are consistent with the general assumption that in birds, females disperse or move further than males during the juvenile dispersal period. Sex-biased juvenile dispersal has been attributed to differences in mating systems with monogamous species showing female-biased juvenile or natal dispersal distance [32]. The Black-shouldered Kite is not a strictly monogamous species. Although it is common that the breeding pairs stay together after the nestlings fledged and complete the post-fledgling dependence period, our field study has shown that it is not uncommon that the female leaves its mate at the care of the chicks while it attempts breeding again with another mate within the same breeding season [50]. Therefore, the mating system alone would not explain the female-biased juvenile dispersal observed in this bird of prey. However, no differences were found between males and females in natal dispersal distances. It is not known if the movement during the juvenile dispersal did not reflect the site were individuals recruited as territorial pairs from the first time or that the sub-sample of individuals used in this study to analyze natal dispersal distance was not large enough to detect these differences.

Under a demographic and conservation perspective, it is important to know why some individuals moved from home areas further away than others [63]. In this study, nestlings hatched from good quality territories disperse shorter distances than those nestlings hatched in low quality territories. The number of years a given territory is occupied is a good proxy of habitat quality [57]. It is expected that those nestlings from good quality territories are those individuals that started the juvenile dispersal period in good body condition and being more competitive than counterpart hatched from low quality territories. Therefore, this result is in line with the predictions under the resource from competition hypothesis. The rank order within the brood has been found to affect the body condition of nestlings. For example, this has been described in Common Kestrel (*Falco tinnunculus*) [64], and other bird species [65,66,67,68]. In our study area, there is a large difference in size between nestlings of the same brood, with the first nestling within the brood sometimes doubling in size the last nestling of the brood (see Figure 1a). Thus, it is expected opposing effects of brood rank order on natal dispersal distance depending on whether the first ranked nestling stayed closer (e.g., predicted by the CRH) or dispersed further from natal areas (e.g., predicted by the WH). Interestingly, a close to significant effect of brood rank order on natal dispersal distance was found. Specifically, first nestling within the brood dispersed a shorter distance than the second or third nestling in the brood rank order. Therefore, this result provides further evidence for the CRH.

Under CRH, it could also be expected that nestlings hatched earlier during the breeding season are more competitive compared to nestlings hatched later. However, no significant effect was found for laying date on natal dispersal distance providing no further support to the resources for competition hypothesis. Many studies on natal dispersal have shown a correlation between hatching date and the probability of recruiting to the natal area or the distance travelled between the place of birth and the site selected to breed for the first time. For instance, the probability to recruit to the natal population is higher for nestlings hatched early in the breeding season in Sparrowhawk (*Accipiter nisus*) or Great Tits (*Parus major*) [69,70]. Laying date is correlated with productivity and habitat quality in most studies on avian biology [71,72]. Therefore, those individuals hatched early in the season came from good quality territories and/or good quality parents and have a higher competitive ability than individuals from late-breeding pairs [23,24,73]. Therefore, the lack of covariation of laying date with natal dispersal distance does not support the RCH.

Results from this work also show that dispersal parameters of a potential colonizer on the move are comparable to those of other small raptorial birds in the same area, such as Lesser Kestrels (*Falco naumanni*) [30]. The Black-shouldered Kite is considered a relatively recent natural colonizer from Africa, where it is, however, abundant and it has a widespread distribution [33,34,35]. In recent years, other African species have been colonizing the Iberian Peninsula. For example, an increasing number of individuals belonging to the African avifauna are sighted crossing the Strait of Gibraltar. This is the case of raptors, such as, the Rüppell’s Vulture (*Gyps rueppellii*) [74]. or the Long-legged Buzzard (*Buteo rufinus*), that recently have been documented as new breeding species for the European avifauna [75,76]. We have shown that Black-shouldered Kites disperse preferably to the North, even though the prevailing winds are South or South-West in our study area. This tendency to disperse to the north may have facilitated the colonization of this species in Europe helped by recent climate warming and/or land use change [54]. The Black-shouldered Kites have settled as one more sedentary raptorial species in at least our study area in southwestern Spain and Portugal and are expanding north where it has been recently cited as a breeding species in France [47]. This species may also benefit because it shows a more prolonged breeding season than any other bird of prey [49]. This is, however, consistent with their dependence on high rodent densities for breeding [55], and the fact that they seem to track rodent population outbreaks quite well in their original range quarters in Africa and elsewhere [77].

In conclusion, juvenile dispersal was described in a species of a poorly studied genus of birds of prey. This general lack of information is surprising given the widespread distribution of *Elanus* Kites (four species) in the temperate areas of all continents except Antarctica [48,77], and the fact that they are the only extant representatives of an ancient and basal group in the phylogeny of the Accipitriformes [78]. As in most bird species, females dispersed further than males. Some evidence was also found in favor of the CRH and no evidence supporting the WH on this species. However, it should be highlight that sample size was small and based mainly on wing-tags marking that might overestimate short-distance and underestimate long-distance movements, respectively. Future studies should increase sample of monitored individuals and use a global positioning system device (GPS) to review these hypotheses in more detail. Furthermore, the recorded events of juvenile dispersal in our population had a significant northernly component, consistent with the northern expansion of the species in Iberia. The information provided should be of great importance because it is a parameter required for some demographic studies; such as, population viability analyses (i.e., PVA) that incorporate dispersal data and thus it is of interest for future demographic and also behavioral studies aimed at preserving the Black-shouldered Kite in Europe, where it is one of the rarest raptors, and to advance research on the mechanisms explaining juvenile dispersal in Aves Class and perhaps other taxa.

## Figures and Tables

**Figure 1 animals-12-02070-f001:**
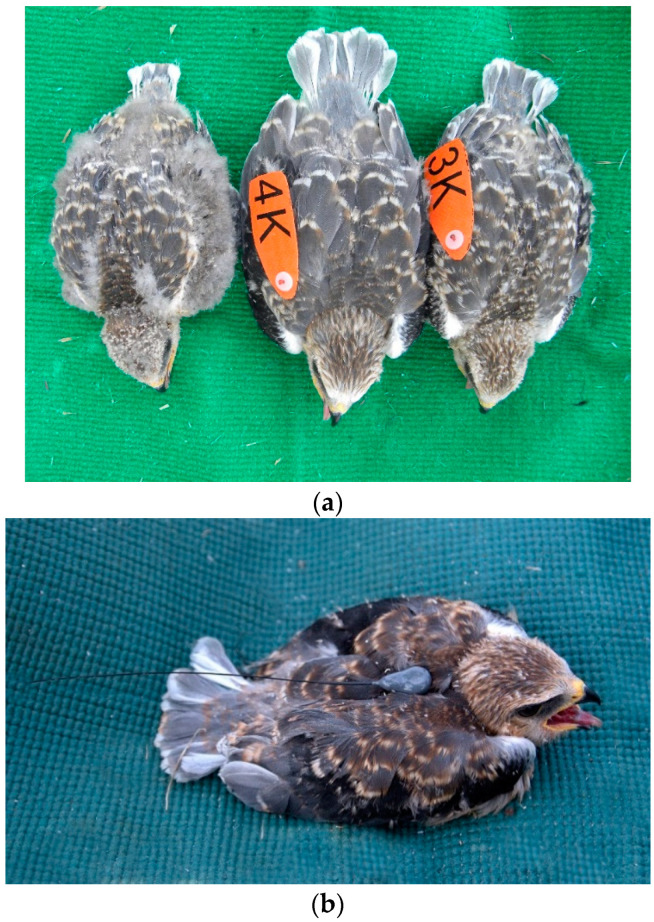
The photographs depicted (**a**) a brood of three nestlings with two of them wearing wing tags on the right wing; (**b**) a nestling equipped with an emisor.

**Figure 2 animals-12-02070-f002:**
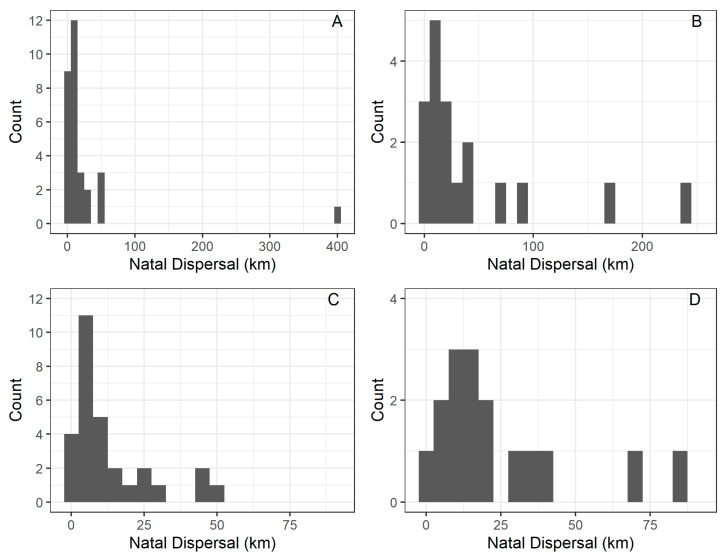
The top panels depicted the juvenile dispersal distance (km) of male (**A**) and female (**B**) Black-shouldered Kite (*Elanus caeruleus*) studied in southwestern Spain. The bottom panels depicted the juvenile dispersal distance (km) of male (**C**) and female (**D**) after excluding one male and two females that dispersed further than 100 km from their natal site.

**Figure 3 animals-12-02070-f003:**
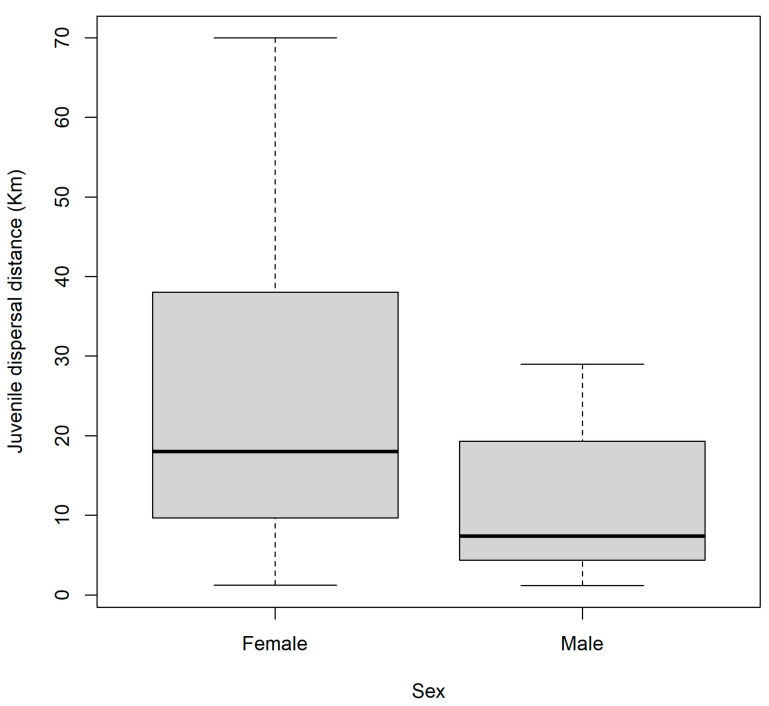
Boxplot for juvenile dispersal distance (km) of male and female Black-shouldered Kites (*Elanus caeruleus*) monitored in southwestern Spain. Horizontal lines = median and boxes are inter-quartile range. Sample size is 30 males and 17 females.

**Figure 4 animals-12-02070-f004:**
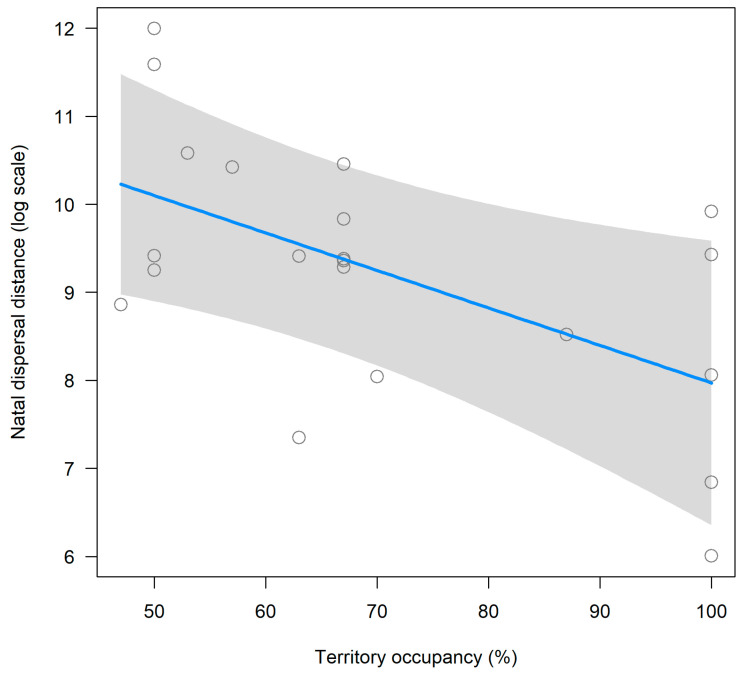
Partial residual plot of natal dispersal distance (km) in relation to territory occupation rate of Black-shouldered Kites (*Elanus caeruleus*) studied in southwestern Spain. The line is the prediction of territory occupation rate on natal dispersal distance while maintaining the other predictor in the model on their mean values. 95% CI are shaded in grey color.

**Figure 5 animals-12-02070-f005:**
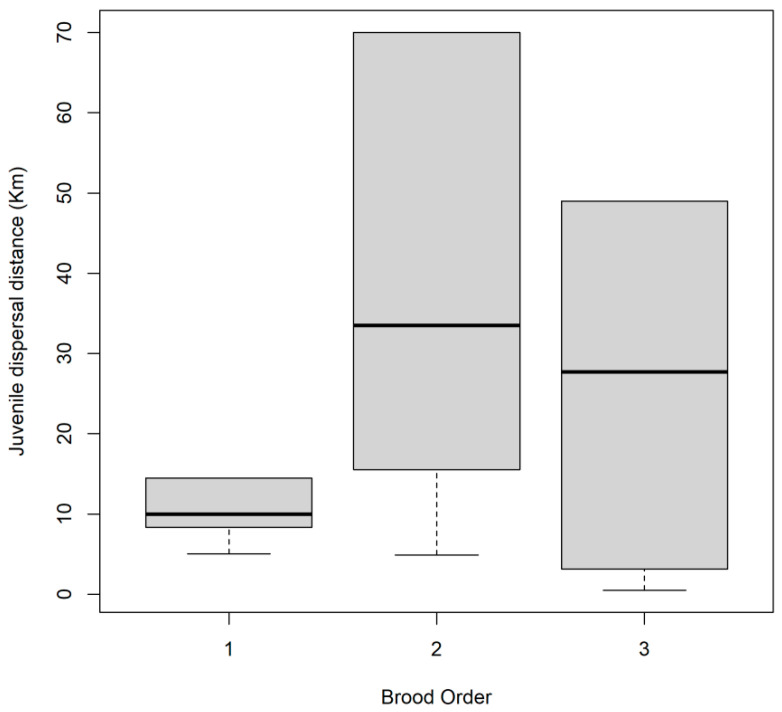
Boxplot for natal dispersal distance (km) according to brood rank order of Black-shouldered Kites (*Elanus caeruleus*) monitored in southwestern Spain. Horizontal lines = median and boxes are inter-quartile range. Sample size is nine, six and six first, second and third-order hatched nestlings within the brood, respectively.

**Table 1 animals-12-02070-t001:** Results from a reduced model selected from a full general linear model (GLM) analyzing natal dispersal distance of Black-shouldered Kites (*Elanus caeruleus*) in southwestern Spain.

Predictor	Estimate	SE	*t*-Value	*p*-Value
Intercept	9.078	0.463	19.58	<0.001
Brood Order (Second)	1.356	0.717	1.89	0.07
Brood Order (Third)	0.955	0.770	1.24	0.23
Territory Occupancy (%)	−1.467	0.653	−2.24	0.038

## Data Availability

Data available on request due to privacy/ethical restrictions.

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
