# Peer review of "Out of Africa: Juvenile Dispersal of Black-Shouldered Kites in the Emerging European Population"

_animals, 2022, doi:10.3390/ani12162070_

Round 1
Reviewer 1 Report
The study is really interesting and has been well structured. Nevertheless, there are a few things that should be improved before publishing the paper.
First of all, the font size changes among the text, please put the size that the editorial recommends. Also, the indentation at the beginning of paragraphs varies in L. 165, 290, 305, 515, 330, and 384. The manuscript must be the most homogeneous possible.
In English, common names are written in lowercase unless the common name contains the proper name of a scientist (eg Bonelli's eagle) or the name of a geographic region or epithet (eg Egyptian vulture). You should review the text and correct all the black-shouldered kite, please. Also, you should correct:
- L. 340: common kestrel
- L. 357: sparrowhawk and great tits
- L. 370: Rüppell's vulture
- L. 371: long-legged buzzard
- L 386: kites
Another recommendation in scientific texts is to use impersonal forms to redact the manuscript. The text mixes both forms: personal and impersonal. In my opinion, the personal form seems to be a bit arrogant, so my recommendation is to change to the impersonal form.
Other points that may be improved are:
- L. 18: please write Elanus caeruleus in brackets. As you mention here the scientific name, it is not mandatory to repeat it after, but you can write one more time in the introduction, the first time you mention the black-shouldered kite (L. 93). Please, remove the scientific name from L. 117.
- L. 82: please change polygonous to polygamous.
- L. 84: please change nestlings to a synonym valid for mammals.
- L. 88: please correct Nonetheless.
- L. 100-114: this part about the discrimination between CRH and WH fit better in Methods...
- L. 117-136: this part about the biology and behavior of black-shouldered kite should be in the introduction, before the aim of the study.
- L. 148-151: you describe the tarsal measurements but you don't specify why you did this technique. I suppose it is to differentiate females and males.
- L.157: why ribbon is in italics?
- L. 174-175: "The mean age of the resighting..." Those data are results of your work, so they should be removed from "Methods".
- L. 192: it seems to be a double space there...
- L. 204-212: again, those data are results of your work, so they should be removed from "Methods".
- L. 216: you write about a statistically significant difference, but you don't fix the limit to consider that difference as statistically significant. Is it 0.1, 0.05 or 0.01? Please include this information.
- L. 243-244: cardinal points should be written with capital letter in English. Please correct here and in L. 373-374 and 378.
- L. 244: "This direction in juvenil dispersal..." which one? Please specify.
- L. 251-252: "As observed in most bird species...". This construction is for Discussion. In Results, you should expose the results without comparatives with other studies or conclusions.
- L. 264: Table 1 appears displaced. Please move it to a central position and put the font size in the same font size as the text.
- L. 277: please put Elanus caeruleus in italics. Do the same in L. 280, 284 and 287.
- L. 332 and 335: I think nestling should be in plural...
- L. 342: author personal observations should not be included in a scientific text, please remove your opinion.
- L. 349: it seems to be a double space there...
- L. 359-361: you repeat twice Therefore... please reformulate.
- L. 366: please add the scientific name of lesser kestrels in brackets.
- L. 371: please put Gyps ruppellii in brackets.
- L. 386: please write Elanus in italicsand kites in lowercases.
- L. 395-396: please correct Aves Class
Reviewer 2 Report
In this study, Rivera et al. assessed juvenile dispersal of the Black-shouldered Kites, a poorly studied small raptor located in different regions of the world including Europe. The authors monitored the southwestern Spain population in order to provide valuable data that can be used for future management and conservation of the species.
The manuscript is overall well-written and the methods and results are clearly presented.
TITLE: informative and attractive
ABSTRACT
The abstract successfully states the goals and results of the study.
Line 26: I do not see where the two hypotheses are stated in the abstract, they were mentioned only when the authors presented their results, so the authors may want to consider revising the abstract to include one or two sentences summarizing the methods and one sentence addressing their hypotheses.
INTRODUCTION
Line 51: The two paragraphs between lines 51-77 could be shortened into a single paragraph on the literature of juvenile dispersal.
Line 99: The paragraph from the line 99 to 114 could be restructured and separated into two paragraphs. The authors shoud first present the studded species and the problematic in a first paragraph and then summarize the objectives of the study in a second paragraph
Line 117: The authors should added sufficient justification for why the studded species represents a model for monitoring juvenile dispersal. In particular, it would be helpful for the authors to move the general introduction (lines 117-136) to the specific introduction of the studded species (two last paragraphs of the introduction) in order to justify for why the Black-shouldered Kites is the focus of this study. The hypothesis needs to be significantly clarified and/or expanded upon.
METHODS
Line 115: The methods are well designed and well presented in the manuscript. However, a map of the study area showing the collection localities would help the reader better understand the dispersal distances and the ability of individuals to move.
Line 115: The authors should address sufficient justifications of the selected statistical analyses to allow readers to better understand how much more appropriate these methods are than others. For instance, why the authors used the Ordinary Least Square regression analyses on the juvenile dispersal distance across sexes or why they selected GLM to study the variation in natal dispersal’ distance instead to GAM which is more flexible for instance… Please clarify.
RESULTS
Line 236: The results section is quite short. That’s not a problem, but I think the authors could explain a bit more the differences they found between the dispersal distances across males and females for instance.
Line 235: The authors should remove the subtitle ‘3.1. Juvenile dispersal’ as there is only one title in the results section; Juvenile dispersal is the title of your paper thus unnecessary to be presented here.
DISCUSSION
Line 301: The sentences “However, it should bear in mind that these results comes from…” lines (301-304)” and “We have monitored breeding pairs of Black-shouldered Kites in approximately 2.800 km2…” lines (137 to 139) are confusing for me. Why the authors monitored only few numbers of individuals in a wider interval of time? They suggest that “future studies will be needed using a larger sample size and new predictors…” lines (201-203). How much time could be sufficient to have larger sample size? The authors should give more precisions and address all aspects that methodologically limit their studies including the low sample size which could affect the statistical analyses.
Line 383: In the conclusion section, lines (383-392), the authors have simply summarized their results instead concluding the paper; this section should be rewritten to focus on the final outcome of the research instead to simply condense the results.
Line 392: The last sentence of the conclusion is better but, there should be more specific explanation drawn from your analyses showing how your results can be valuable for future demographic and behavioral studies.
Line 392: In the ABSTRACT section, the authors concluded that “The information provided by these crucial demographic parameters will be used for the elaboration of future conservation plans for the management…”, lines (31-33), these aspects need to be developed a little more in the conclusion of the paper to allow the reader to better understand how your data are valuable for future management and conservation plans of your study species.
